# Understanding Church-Led Adolescent and Youth Sexual Reproductive Health (AYSRH) Interventions Within the Framework of Church Beliefs and Practices in South Africa: A Qualitative Study

**DOI:** 10.3390/healthcare13080907

**Published:** 2025-04-15

**Authors:** Vhumani Magezi, Jaco Hoffman, George W. Leeson

**Affiliations:** 1Optentia Research Unit, Faculty of Humanities, North-West University, Vanderbijlpark 1900, South Africa; jaco.hoffman@nwu.ac.za; 2Oxford Institute of Population Ageing, University of Oxford, Oxford OX2 6PR, UK; 3Oxford Institute of Population Ageing, Kellogg College, University of Oxford, Oxford OX2 6PN, UK; george.leeson@ageing.ox.ac.uk

**Keywords:** church health assets (CHA), religious health assets (RHA), church health assets in youth sexual reproductive health (YSRH), adolescent and youth sexual reproductive health (AYSRH) in churches, adolescent and youth sexual reproductive health (AYSRH) in South Africa

## Abstract

**Background:** The existing literature often oversimplifies the complex relationship between religion and Adolescent and Youth Sexual and Reproductive Health (AYSRH), particularly regarding church-based interventions. This study aimed to investigate the nature and implementation strategies of church AYSRH programmes within their belief systems to inform effective programme development. **Methodology:** An interpretive descriptive design was employed. Data were collected in the Vaal Triangle region of South Africa (Vanderbijlpark, Vereeniging, and Sasolburg) between August 2019 and February 2020. In-depth interviews were conducted with pastors, government officials, and school principals. Focus group discussions were held with parent and youth church groups alongside youth groups from Technical and Vocational Education and Training (TVET) colleges. Data were analyzed using Atlas.ti v.23. **Results:** Church-based AYSRH interventions primarily emphasize information provision and abstinence until marriage, aligned with prevailing moral values. These interventions are delivered through integration into existing church programmes and collaborations with external stakeholders for specialized areas like contraception. Limitations identified included ineffectiveness, superficiality, impracticality, tensions between religious doctrine and lived realities, a reductionist focus, a singular information-sharing approach, and limited pastor understanding and openness regarding AYSRH. **Conclusions:** Churches possess valuable communication platforms and partnerships that could be leveraged for AYSRH interventions. However, this study highlights a one-sided focus on church teachings and significant tensions between idealized approaches and practical implementation, raising critical questions about the overall efficacy of church-led AYSRH projects. This research validates prior findings on church-based AYSRH interventions while offering nuanced insights and heuristic perspectives for a more comprehensive and less simplistic understanding of church-driven AYSRH services.

## 1. Study Background

The relationship between religion and Adolescent and Youth Sexual and Reproductive Health (AYSRH) is complex. It involves conflicting viewpoints. These differing perspectives can make it hard to deliver effective services. Several studies support this observation [1,2,3,4]. For instance, Abdulla [5] found that religious beliefs strongly influence sexual and reproductive health practices among married couples in rural Kenya. Abstinence, faithfulness, and opposition to contraception are key factors. Religious leaders also have a significant impact on the practices of their followers.

Studies consistently highlight many challenges related to religion and AYSRH. These include moral and religious beliefs. They also involve the stigma associated with sensitive topics. Gender roles and norms play a part, alongside varied interpretations and hierarchies within religious structures. Resistance to change and cultural and societal factors also contribute to these challenges [1,3,6,7]. Religious moral frameworks often emphasize abstinence. This can create tension when discussing premarital sex, contraception, and abortion. Taboos surrounding topics like contraception, abortion, and sexual diversity lead to stigma. This stigma hinders open dialogue and access to healthcare. Traditional gender roles within religious contexts can worsen inequalities, especially for girls and women.

Different interpretations of religious teachings and hierarchical structures further complicate discussions about AYSRH. Resistance to change within religious institutions can slow down the implementation of programmes based on evidence. In many African societies, religion is deeply intertwined with culture. This creates additional obstacles to addressing AYSRH. To overcome these challenges, researchers stress the importance of involving religious leaders and communities as partners. It is crucial to address misconceptions, highlight shared values, and provide information based on evidence [6,7,8,9,10,11,12,13,14]. Respecting religious beliefs while advocating for young people’s rights and well-being is essential for effective AYSRH interventions.

Much research on how religious organizations contribute to AYSRH focuses on religious health assets (RHAs) [14]. Schmid et al. [15] define RHAs as resources within religious entities that can be used for development or public health. Morgan and Ziglio [16] define a health asset as anything that improves health and well-being. RHA research, particularly in Africa, has centred on the African Religious Health Assets Programme (ARHAP). This programme aims to strengthen faith-based health initiatives [9]. Given that Christianity is a large part of the population in sub-Saharan Africa [17], churches are a critical area for investigation.

Notably, the role of religious organizations, especially churches, in Adolescent and Youth Sexual and Reproductive Health (AYSRH) has often been described in general terms. These descriptions rely on assumptions about their potential value rather than on actual evidence. Tomalin et al. [18] note that this approach can overstate the benefits of faith-based organizations (FBOs) in health. There is a significant lack of research based on real church-based AYSRH interventions. Van Bortel et al. [13] emphasize the need for studies that define, theorize, and evaluate the necessity and added value of religious health assets, including church-based AYSRH initiatives. Olivier and Wodon [19] accurately describe the situation as having inconsistent data. It also involves information gaps, different ways of looking at things, and often conflicting opinions and goals. This lack of consistent and strong supporting evidence hinders effective policy recommendations.

To move beyond theoretical claims and understand practical realities, empirical research is crucial. This study aimed to identify and describe the nature of AYSRH services within churches. It recognizes the importance of understanding the perspectives of different people involved. It is essential to understand exact practices, beliefs, and views to ensure that interventions connect with the local community. Authority figures, such as parents and caregivers influenced by local religious beliefs, shape how adolescents understand SRH [19,20,21]. Therefore, it is important to examine church-based AYSRH interventions from the viewpoints of church leaders (pastors), parents, young people, and other stakeholders, including schools and government officials. Church leaders may believe that their activities, which align with their faith, are meaningful AYSRH interventions. However, young people may see them differently. For example, the emphasis on abstinence might seem unrealistic. Conflicting messages about contraception may exist between the church and school settings.

Golo et al. [22] highlight the constant negotiation between knowledge learned at home, church, and school and knowledge gained from peers and other influences. This shows that even well-intentioned interventions can be ignored or rejected. Van Bortel et al. [13] point out the unclear nature of religious health-based services. There are unclear definitions of what counts as AYSRH services and who decides these definitions. As Karam et al. [11] and others note, the exact nature and impact of faith’s role in healthcare are not well understood. Cochrane [9] emphasizes the need to explore the internal aspects of religion. This means understanding the motivations, commitments, attitudes, and actions of faith-based communities based on their own understanding and beliefs.

While discussions about RHAs are often theoretical, many studies examine practical church-based AYSRH interventions [23]. Building on the work of Cochrane [14] and others, research and programmes like the HOPE Project (SAFFI), CCMP (Malawi), Faith to Action Initiative (Uganda), church-based Adolescent Sexual and Reproductive Health Project (Kenya), and PACT Faith-Based Initiative (Nigeria) explore AYSRH within congregations [24,25]. This study aims to further contribute to understanding church interventions at the congregational level. The study explores AYSRH interventions implemented by churches where this programme fits with their faith beliefs and values. The goal is to understand the interventions that churches themselves consider relevant based on their own understanding and worldview.

## 2. Research Question and Aim

This article investigates how churches in the Vaal region (South Africa) address Adolescent and Youth Sexual and Reproductive Health (AYSRH). It seeks to answer the following research questions:What are the perceived interventions or services being implemented by churches within the framework of their beliefs and practices?How are these interventions implemented?

By exploring these questions, the study aims to understand how churches are currently involved in AYSRH and identify ways to leverage their interventions for more effective youth health outcomes.

## 3. Methodology

### 3.1. The Research Design and Process Followed

This study employed a qualitative research approach. It utilized purposive sampling within a pragmatic paradigm and an interpretive description research design, as described by Thorne [26]. The first author collected data over seven months, from August 2019 to February 2020. The research took place in the Vaal Triangle region of South Africa, specifically in Vanderbijlpark (VDBK), Vereeniging (VR), and Sasolburg (Sasol).

The sample population comprised key stakeholders pertinent to Adolescent and Youth Sexual and Reproductive Health (AYSRH). Specifically, it encompassed ecclesiastical leaders (pastors) responsible for the implementation of AYSRH interventions within religious settings, parental figures from church congregations who possess perspectives and expectations regarding church-based AYSRH initiatives, educational administrators from local schools who observe and engage with youth on AYSRH issues within the school environment and who are affiliated with local churches, young congregants from the community’s churches, and young people attending tertiary institutions within the target communities who possess exposure to both collegiate and community-related issues, while also participating in church activities. This diverse participant selection was crucial to comprehensively capture the multifaceted perspectives of individuals involved in AYSRH. Discerning local practices, beliefs, and viewpoints is important to ensure the contextual relevance of interventions. Consequently, this study incorporated the perspectives of authoritative figures, such as parents and caregivers, whose influence, shaped by local religious tenets, impacts adolescents’ comprehension of SRH, alongside the direct viewpoints of young people themselves.

Drawing from the above purposive sample, data collection involved in-depth interviews with eleven pastors, three school principals, and two government officials. Additionally, focus group discussions (FGDs) were conducted. There were three parent groups and three youth groups from churches. Two FGDs were also held with students from two Technical Vocational Education and Training (TVET) colleges in Sedibeng. Data analysis followed a qualitative approach aligned with interpretive description, as advocated by Creswell [27]. The qualitative data analysis software Atlas.ti was used.

The selection of churches and participants was carefully considered. Eleven churches were chosen from an initial pool of twenty-one to ensure racial diversity. Churches were selected based on their approach to AYSRH interventions (open, conservative, or Pentecostal), their size (at least 100 members with a significant youth population aged 18–24), the presence of families with young people, and the pastor’s willingness to participate. This selection resulted in interviews with the eleven pastors of the chosen churches. Of the eleven churches, four had a mixed racial composition, primarily White, with other racial groups represented. Seven churches were predominantly Black African. Two churches were identified as open to AYSRH, four were conservative, and five were Pentecostal and conservative.

Three focus group discussions were conducted with parents and young people from the selected churches. Participants were required to be regular church attendees. Conducting FGDs with parents and young people from the same churches allowed for a comparison of perspectives within the same church environment. Young people participating in the FGDs were between 18 and 24 years old, which is a group largely comprising those completing high school and attending college.

Two focus group discussions were also held with students aged 18–24 at two Technical Vocational Education and Training colleges in Sedibeng (one in Vanderbijlpark and another in Vereeniging). These colleges were chosen to gather broader views on church-related AYSRH. Three high school principals were interviewed and selected based on their experience in mixed schools within the Vaal region. Two government officials from the District Department of Social Development and the Department of Health were also interviewed. In total, seventeen interviews and eight focus group discussions were conducted.

The interviews and group discussions lasted between 60 and 80 min, depending on the responses. This average length was chosen to avoid overstretching participants and to stay within the recommended 60–90 min timeframe. Through the interviews, the author gathered detailed information on perspectives regarding church AYSRH assets alongside activities, capabilities, skills, resources, links, associations, organizations, and institutions within communities that could be used for AYSRH. The interviews were audio-recorded with participant consent, and detailed notes were also taken.

Ethical approval for the study was granted by the North-West University Basic and Social Sciences Research Ethics Committee (BaSSREC) (approval number NWU-00879-19-S7). Permission to interview pastors and conduct FGDs with churches in the Vaal was obtained from the Greater Vaal Pastoral Forum, which is the umbrella body of pastors. Before starting the research, a discussion was held with the pastor leadership in the Vaal to acknowledge the sensitivity of AYSRH discussions within churches. This awareness prompted the first author to approach the subject with sensitivity. It was noted that pastors held diverse and sometimes conflicting views. Therefore, caution was exercised from the beginning. Furthermore, all stages of the study, from defining the question to publishing the results, followed standard ethical principles [28]. These principles included obtaining permission and informed consent, discussing benefits and potential risks, addressing deception, ensuring privacy and confidentiality, maintaining accuracy, and properly storing records [29]. Participants signed an informed consent form.

### 3.2. Data Analysis and Analysis Credibility

The collected data were analyzed using a qualitative data analysis approach [27]. Given the interpretive descriptive nature of the study, the analysis followed guidelines from this approach. Thorne [26] suggests that analyzing data in an interpretive descriptive design focuses on the phenomenon and the context of the practice field. This involves coding to organize data into similarities and differences, identifying patterns and relationships, and making associations using techniques like constant comparative analysis. These patterns are then transformed into findings, which may also include recommendations for improvement, policy changes, educational content, or how practitioners can be more sensitive and informed.

This guidance from Thorne [26] was integrated into the primary analysis method used: thematic content analysis (TCA). The first author employed TCA, which involves identifying qualitative data themes and exploring how these themes connect within and across datasets. TCA followed the six stages outlined by Braun and Clarke [30]: familiarization with the data, generating initial codes, searching for themes, reviewing themes, defining and naming themes, and producing a visual representation and a written report. Thematic coding was used to sort and organize data, identify patterns, and understand relationships.

These patterns were examined to establish links to the research question and relate them to the study’s objective. Recognizing the fluidity of qualitative data, Thorne [26] advised that codes should be flexible and serve as signifiers and language references during the sorting process. Therefore, data were read, reread, interpreted, and reinterpreted iteratively to capture nuances in the information provided by the pastors and to establish a credible and transparent analysis. Participants in interviews and FGDs were assigned pseudonyms, as shown in Table 1.

In line with the interpretive descriptive approach, rigour in this qualitative research was ensured through four processes [26]. First, epistemological integrity was maintained by clearly stating the underlying assumptions about knowledge and the corresponding methodological and research processes. The study’s pragmatic paradigm and interpretive description design were presented as the foundational frameworks for investigating AYSRH and churches. The influence of these approaches on the entire research process (paradigm—design—methodology—data collection—analysis—reporting) was consistently argued, demonstrating a clear, logical flow.

The second principle, representative credibility, ensured that the study’s claims aligned with the sampling method. It emphasizes careful knowledge claims, the value of prolonged engagement, and the importance of data source triangulation. To ensure credibility, this study involved a seven-month data collection period (August 2019–February 2020). Data were gathered from six different sources to allow for triangulation and the comparison of perceptions and views. The study’s focus on AYSRH and church assets was informed by a robust literature review and the first researcher’s extensive practical experience, providing a strong rationale for the research question. The pragmatic principle of “research results that work” in real life guided the assessment of the study’s knowledge claims regarding AYSRH.

The third principle, analytic logic, requires that the reasoning behind the interpretations is logical from the beginning, with clear evidence provided in the report. Decision-making processes should be accessible, and an audit trail should be maintained. Verbatim accounts and illustrative materials must be included to provide context. This study includes detailed explanations of the research processes and decisions made to clarify the underlying logic and rationale. The research reports (manuscripts) are written in clear academic language. Furthermore, the reports contain verbatim quotes that provide evidence of the source information and the voices of the participants, serving as the basis for the findings and conclusions.

The fourth principle, interpretive authority, ensures the reader’s trust in the researcher’s interpretations. It involves establishing some form of external validation of the researcher’s perspective, for example, through sufficient information. This also includes mechanisms to check interpretations against participants’ views. In this study, the methodology includes a section on the first researcher’s positionality and role, outlining their experience, competencies, viewpoints, and a commitment to a “stand back approach” to allow the data to speak for themself. The author’s commentary and interpretations closely align with the data presented throughout the Results and Discussion sections, enabling readers to evaluate the interpretations against the provided information.

In addition to these approaches, the analysis incorporated negative cases and the constant comparison of responses across participants to gain a comprehensive understanding of the research phenomenon [31]. Constant comparison was used to assess the consistency and accuracy of the interpretations provided by participants during discussions.

### 3.3. Role of the Research

The primary investigator, who conducted the fieldwork and subsequent data analysis, possesses dual qualifications as a trained (Ph.D.) and practising practical theologian, complemented by training in health development (MA). Theologically, their perspective aligns with a progressive conservative stance, with a specific focus on public practical theology. Their engagement in this study was as a senior researcher, bringing over two decades of experience in HIV and sexual and reproductive health (SRH) interventions, accrued through work in non-governmental organizations (NGOs) and the execution of numerous AYSRH studies for international bodies such as UNESCO and UNFPA, among others. This extensive experience had a dual impact on the research. On the one hand, it provided a valuable framework for the author’s reflexive engagement and analysis of the subject matter, informed by phenomenological, theoretical, and pragmatic considerations, thereby enriching the analysis and contributing nuanced insights. Conversely, this prior experience introduced a potential risk of analytical bias, wherein the author might prematurely draw conclusions based on previous work without rigorous in-depth analysis. To address this, the author adopted a reflexive approach, acknowledging the positive contribution of their experience to the analytical insights. To mitigate the potential for analytical bias throughout the research, the author consciously refrained from premature conclusions and prioritized allowing the data to guide the interpretation. This was operationalized through the provision of rich descriptive accounts and verbatim quotations, which are extensively incorporated within the Findings section to facilitate reader validation of the analysis. Furthermore, to counter potential reader scepticism, it is pertinent to note the author’s consistent inclination towards a pragmatic methodological approach within practical theology. This pragmatic paradigm, coupled with the interpretive and descriptive research design, emphasizes a logically traceable research description process, serving as a methodological counterbalance for the reader. The study provides a detailed exposition of the logical processes and steps undertaken, including the explicit articulation of assumptions. Consequently, the overall impact of the first author’s experience within this research is predominantly assessed as positive.

The first author’s foundational training in theology and international development exhibited limited exposure to in-depth sociological empirical methodologies. This deficiency was addressed through advanced training in qualitative methods aimed at refreshing and enhancing their understanding of key concepts and the complexities inherent in conducting empirical social science research. Consequently, refresher training encompassing the entire research process was undertaken to strengthen the study’s methodological rigour. This investment proved particularly beneficial with the implementation of a novel interpretive descriptive research design. Thus, the author demonstrated both competence and confidence in executing the research.

A notable challenge associated with interpretive descriptive design lies in its less maturely developed methodological framework despite its recognized utility. This characteristic carries the potential risk of findings being perceived as lacking in methodological rigour. To ensure the highest standards of analysis and to establish the academic credibility of the research findings, the author compensated for this design limitation by employing a comprehensive approach to data analysis credibility. This involved the adoption of analysis validity and credibility strategies drawn from diverse methodological frameworks rather than adhering solely to the guidelines typically suggested by Thorne [26].

In summary, the positionality of the first author within this study is acknowledged, encompassing both their inherent strengths and potential limitations. While their extensive experience enriched the research through meticulous and careful management of the research process, a lack of critical engagement with this experience could have potentially compromised the study’s findings. Therefore, it is our understanding that the first author’s contribution to this research should be appreciated, particularly in their effective utilization of the positive aspects of their background.

## 4. Findings

This study investigated AYSRH interventions offered by churches, examining how they are shaped by religious beliefs and practices, with the goal of understanding and leveraging these interventions for enhanced effectiveness.

### 4.1. Church AYSRH Interventions

A thematic map (Figure 1) summarizes the seven types of interventions identified, which are followed by detailed description and discussion.

#### 4.1.1. Partnerships for Mutual AYSRH Services with Stakeholders

Churches form partnerships with different stakeholders surrounding them to provide different types of AYSRH services. These partnerships take four forms.

The first partnership entails churches partnering with non-governmental organizations (NGOs). Some examples of NGOs mentioned are Men on Track (P1), Heartline South Africa and Silver Rings (P5), and Siyafundisa (P11). In this partnership, churches provide the meeting space (venue), and the young people (participants) attend the AYSRH sessions, while the NGOs provide the various types of interventions or services that pastors lack the capacity to tackle and topics that pastors deem to be too controversial to address. The interventions include teaching boys about aspects of manhood and respecting girls, teaching girls about womanhood and respecting boys. The teaching is usually performed by the NGO’s peer-trained staff and experienced senior staff. The partnerships enable the churches and church leaders to overcome their capacity limitations on some AYSRH issues, and they ensure that controversial and taboo topics are handled by an outsider to preserve the pastors’ respectability and avoid controversy in the church. One pastor clearly described what other pastors had also said as follows:

*My church partners with NGOs to provide youth with SRH programmes although some programmes such as teaching young people to condomise are in conflict with church teachings. My church partners with the NGO, such as Matlafala that deal with youth. They run youth programmes on sexual reproductive health issues. They also do sessions for boys and girls. Boys are taught about manhood, sexual reproductive health issues, how to respect girls and elders. The programme sometimes brings young boys and girls together to empower them on anything that has to do with youth. The facilitators of the Matlafala programme are well trained on youth issues and are able to equip the youth with reliable information on public health services*.(P9)

However, while the NGO addresses controversial AYSRH topics, tension and conflict between them and the church is still experienced. The same pastor (P9) explained that “*some of the teachings such as condomising are in conflict with church teachings and beliefs. This is uncomfortable among the members as the children give feedback to their parents about the teachings*”.

The second form of partnership is similar to NGO partnerships but slightly different. In this partnership, churches utilize the services of professional people, such as doctors or nurses who belong to the church, from nearby government or private health facilities, to conduct SRH information sessions. One pastor asserted the following:

*Our church collaborates with health professionals by way of bringing on board extra knowledge that the pastors don’t have from their theological college training. Here, I mean, sexual reproductive health is not even part of the pastor’s curriculum in theological colleges, so there is a need for the churches to partner with specialists on sexual reproductive health by inviting these specialists to come and teach our youth at church*.(P4)

Referring to partnering with government health facilities, another pastor added the following:

*The Department of Health officials assist us to responsibly teach about abstinence and the repercussions of risky sexual activity for the youth, they also have sessions with parents, giving them skills on how to speak to their children about SHR. It’s very difficult for Christian parents to talk to their children about SHR because they like to spiritualise things and like to focus more on victims than on prevention. Christian parents talk to their children about SHR when something has gone wrong*.(P9)

The advantage of this partnership is that even though controversial issues are accepted grudgingly, the services and activities are facilitated by knowledgeable people in a sensitive manner, which lowers conflict and tension. For instance, one pastor stated the following:

*The discussions are facilitated by experts within their age groups. The job of the pastors partnering with the Matlafala programme is to open these youth gatherings with a prayer and then encourage the youth with the Word of God that will be relevant to the discussion on that day*.(P8)

The third form of partnership is the reciprocal relationship that churches and their pastors establish with stakeholders working with young people, such as schools. The partnership involves pastors addressing school learners on some AYSRH topics while teachers provide and complement AYSRH information in churches. This relationship enables some stakeholders to complement the churches’ AYSRH information by addressing topics that churches are uncomfortable dealing with. For instance, one school leader stated the following:

*Churches struggle to discuss [sex] and cannot handle [sex discussions] because of its church practices and beliefs on sex. The church encourages abstinence. And the church will say you should not have sex before marriage, because it is a sin according to the Bible. Even though it is sin, most youth want to experience sex, and they are doing it just to experience it. They will ask what is wrong if I have it before marriage. Viewing it as sin does not discourage many of them from indulging in sex. But here at the school, we tell them that if you do it, you are going to contract diseases, you are going to have a baby and at the end, you are not going to attend school fully and you are not going to fend for yourself in the future. And if the person who has impregnated you does not marry you, it means you will raise the child alone and this has serious psychological implications on the girl*.(PT2)

One pastor explained the partnership between schools and pastors. He stated that the following:

*SRH is a multidisciplinary issue. The church can only deal with the youth problems in the church, yet there are other youth outside the church, i.e., in schools that require support for SRH. So, there is need to partner with other stakeholders like schools*.(P9)

The third area of partnership can be related to the fourth area, which describes churches and church leaders performing a public role related to AYSRH. Rather than focusing on church members only, church leaders perform a public role among public institutions such as schools and in public activities where AYSRH issues are discussed. “*As a pastor, I am sometimes invited by schools to talk to the young people about SRH issues. The discussions vary, but they include issues of morality and living a responsible life as a young person*” *(P1)*.

In addition to partnerships formed by churches to provide various AYSRH services, churches also conduct their own specific workshops. The focus of the church workshops is to discuss sensitive AYSRH issues based on a “biblical SRH package” as seen in the Bible. The discussions aim to develop knowledge so that church members understand ways to deal with AYSRH matters, “especially sex issues”. This is intended to ensure that church messages are appropriate. However, people do not discuss sex issues openly during the workshops. In a focus group discussion, one parent explained that the following:

*In our church, the pastor periodically conducts seminars on the biblical notion of sex. This is an environment where young people can ask questions, though the discussions are not detailed in the church due to lack [of] openness on sexual discussion*.(PC2)

During workshops, and especially workshops held with the youth, young people are confronted with the harsh realities of sexual irresponsibility. Sometimes, videos are shown, and interviews are conducted with (homeless) people on the street to indicate examples of the negative effects of engaging in an early sexual debut, failure to abstain from sex until one is married, and also the consequences of not engaging in safe sex. One pastor described the activity at his church as follows:

*At my church, I am using the testimonies of the homeless people to impact the young people’s lives. We do this almost every Sunday, where we go out in [the] streets with the youth to minister to the homeless people. Here, we ask the homeless young girls and boys about their stories. All the stories of these homeless people usually show us that things didn’t start bad in their lives, but they made wrong choices. Thus, by exposing the youth to the effects or consequences of the life of the homeless people that started well, but are now going bad, the youth learned to behave appropriately in all aspects of life, including sexual matters*.(P7)

Churches also focus on protecting young people from transactional sexual activities by teaching them about good sexual conduct as an act of self-love and self-value and overcoming the pressure of transactional sex. These activities are performed in mixed groups between church and non-church youths from the community. One pastor explains what they emphasize in the AYSRH and sex teachings.

*We run church and community youth teachings about self-value, which means respecting yourself. In these church programmes, we invite non-Christian youths. You are valuable; therefore, take care of yourself, respect your body and don’t give yourself away, preserve yourself for one man or woman. Be sexually pure for your own benefit*.(P6)

#### 4.1.2. Capacity Development of Parents for AYSRH Services and Parental Roles

Churches are involved in developing the capacity of parents to address AYSRH. Capacity development entails building parents’ skills by training them in effective parent–child communication. This ensures that parents’ abilities are enhanced to discuss, advise, and answer their children on different AYSRH issues in a culturally and biblically based manner. One pastor emphatically stated that parents must be equipped to address SHR in their families, and there are activities in the church to address this. She stated that the following:

*We have women and men’s groups that meet regularly. Must they always meet to preach? SRH is also for the parents to know about the topic. Some of the parents are single parents. How does a widowed African father talk to his daughter about her first menstruation without having to send her away to an aunt? The church really needs to do more. So, we hold these discussions in these different discussions in our church*.(P11)

There was also an emphasis on the need to develop and extend parental capacity to discuss AYSRH topics beyond the church environment to the home situation. The home was identified as a critical space for AYSRH information; hence, parents need to be skilled in that regard. One pastor indicated that the following:

*It is important to empower parents to communicate with their children on AYSRH issues at home. People should know that ‘communication, communication and communication’ is important. There should be more conversations about SRH between the church pastor and parents, as well as church elders and parents. Parents should discuss with children like a friend to a friend in the home. This means the church should emphasise biblical truth—help parents to know God’s biblical truth on sexuality, so that parents can initiate biblical conversations with their children, so that children will have conversations with their friends. This is what we do in our church parenting and family discussions*.(P6)

Training parents in the church also extends to identifying children who have no parents in the church and allocating parental caretakers to them. Assigning parents to young people with no parents in church allows these children to receive parental care and guidance on different AYSRH issues from a biblical perspective. In one parent FGD, a participant indicated the following:

*The church has an adoption programme for children that do not have their parents in the church. Children whose parents are not members of the church are adopted by families within the church as a way of raising them within a Christian family context. This provides a platform to teach them about the church’s expectations on sexual matters, which these youth may not receive from their homes that don’t come to the church*.(PC1)

#### 4.1.3. Mentoring and Information Provision on AYSRH

Churches are providing AYSRH mentoring to young people. This mentoring aims to build skills that can help the youth abstain from sex and prepare them for marriage. Church leaders are performing the AYSRH role by focusing on ensuring moral values and chastity are upheld as critical virtues. One pastor explained the mentoring as follows:

*We hold personal conversations with the youth as a form of mentorship and accountability. This encourages healthy conversations on sexual issues. The youth must make themselves vulnerable by approaching the pastor or a church leader and open up about their problem and the leaders will point him to the biblical resource. The leaders make themselves available to the youth to support and encourage them to uphold abstinence. And it takes courage for the youth to open up*.(P6)

However, the pastors acknowledge that mentoring is a difficult task because young people do not always open up or come forward to be assisted. There is discomfort among young people in discussing AYSRH issues with pastors despite the mentoring invitation. As one pastor admits, “*Young people don’t easily open up to pastors. They are uncomfortable to share their secrets with them. It takes courage for the youth to open up to leaders in the church, especially the pastor as a leader (P7)*”.

As a way of ensuring that young people in churches readily access AYSRH information, some pastors open up their churches to provide support as information centres. For example, one pastor stated that the following:

*At our church, we provide information as an information centre, although it’s not a lot. The church provides a youth desk where the youth can get information in the church regarding sexual reproductive health issues. The youth desk makes it easy for the youth to access the information of sexual reproductive health and many more other issues, such drug abuse, and so forth*.(P9)

#### 4.1.4. Individual Advice Counselling and Mentoring of Young People on AYSRH

One-on-one counselling, open discussions, and church spaces are used as mentoring platforms. Some church leaders hold one-on-one meetings and discussions with young people who present with challenges in order to help them. These discussions help young people open up about different related AYSRH challenges. One pastor stated the following:

*I hold conversations with single men as an effort to minimise the damages that I would have noticed. These youths come to me with their problems, i.e., they have made a girl pregnant, so they don’t know what to do. So, I always have conversations to try to help and advise them from the Word of God*.(P8)

Some churches also hold non-directive open discussions with young people in an ad hoc manner. The discussions are performed as part of mentoring discussions, church workshops, youth groups, and parent and child conversations. They are also held as a form of follow-up after identifying signs of a problem from a young person or family. One parent in an FGD stated the following:

*We hold a mentorship programme in our church where older Christian men meet with young boys to talk about their personal growth and sometimes issues of sexuality. Sometimes, they go on a father and son camp where issues of sexuality are part of the agenda. During Youth Day, father and son/mother and daughter issues of sexuality are also discussed. Sex is a taboo in many traditional African families, instead of planning to have sex talk, unless in rare occasions, sex issues are addressed in ad hoc situations*.(PC1)

#### 4.1.5. Workshops on AYSRH (Sexual Issues) and Moral Teaching—Good Sexual Conduct

All church leaders reported that they participate in public youth events, but their focus on AYSRH activities differed. National Youth Month, which culminates in the observance of the June 16 holiday (National Youth Day) and the Day of the African Child, is often used as pastors’ focal days to discuss AYSRH. However, these participations and activities are not properly organized. They are often ad hoc. One pastor specified that the following:

*We do not have our foot on the pedals. Our AYSRH activities are not systematically done or planned as an annual event always. The focus is more on youth during the National Youth Month in June where we try to emphasise AYSRH issues*.(P1)

#### 4.1.6. Church Spaces for AYSRH Material Development: Information Centres and Information Dissemination Through Dedicated Church Services

Churches are involved in church-specific AYSRH programmes like AYSRH material development and material provision to prevent transactional sex. There are AYSRH-dedicated Sundays where AYSRH content is discussed.

Apart from being assisted by NGOs or running usual church youth programmes, some churches also have specifically developed AYSRH-related programmes. Denominations have developed church AYSRH customized material. One pastor reported the following: “*Our church runs a programme called Siyafundisa, which uses a Manual that has an abstinence slant, but also includes life skill topics, such as teenage pregnancy, safe sex, drugs and alcohol*” *(P11)*.

Further to conducting activities under a specific programme, churches conduct dedicated Sunday services to discuss SRH issues, such as “*a Wellness Sunday*” (P11). Experts are invited to address the church on AYSRH-related issues during these Wellness Sundays. During special Sunday services, for example, Youth Sundays or Wellness Sundays, “*specialists or pastors share their knowledge on SRH and guide parents and youth in church*” (P11). Churches also provide material resources to prevent young, poor girls from engaging in transactional sex for survival. In one parent FGD, the participants explained that their church offers the following:

*…food, education, clothes, and shelter to some vulnerable young girls, so that they won’t resort to transactional sex to address their material needs. The church views transactional sex as bad; it violates the churches’ fundamental teaching on sex, i.e., no sex before marriage. Further, it exposes girls to STIs (such as HIV/AIDS) and unplanned pregnancy that can ruin their careers and education*.(PC3)

#### 4.1.7. Church Camps and Indaba to Develop AYSRH Life Skills

Apart from church capacity in the form of buildings and workshops, churches run church youth camps and youth *indaba* (a South African forum to discuss important issues) on youth sex and life skills. The parent FGD at one church stated that they “*hold youth camps where they effectively address some aspects of sexuality*” (PC2). Similarly, an FGD for young people reported going to camps and holding *indabas* where they discuss AYSRH issues. The young people’s FGD stated that “*as a church, we recently had an indaba where professional people were invited to come and address us youth on such sexual issues*” (Y1).

### 4.2. Divergences in Churches’ AYSRH Services

Despite AYSRH services being implemented by churches, four categories of divergences among participants were noted. These divergences are summarized in Figure 2 below and described in the section that follows.

#### 4.2.1. Non-Meaningful and Ineffective Activities

There is a feeling by some pastors, young people, parents and government officials that the churches’ AYSRH services are ineffective and not meaningful, while some churches do not have any such activities at all. Whereas some pastors indicated different forms of AYSRH services being implemented in churches, others held different views. The pastors reported that “*nothing or very little is being done on AYSRH*” (P8), implying that a lack of interventions are deemed meaningful. Similarly, young people in churches alluded to the same experience of viewing the activities as “*not meaningful*” or “*nothing happening*” (Y1–3). One pastor provided a useful summary:

*The role of the church towards contributing to AYSRH at individual, family, community, and government level is to spearhead the conversation. However, from my experience, currently churches have done a bad job because they hardly speak about this issue at church and beyond the church circles. My church has to be blamed as well because this is the first time I am meaningfully having this AYSRH conversation*.(P8)

Another pastor added that churches are not doing much to directly contribute towards AYSRH other than enforcing moral behaviour. He substantiated this by saying the following:

*The teaching on AYSRH in churches happens in sermons where the pastors usually touch the issues of abstinence and being faithful to your partner. There are no specific sermons or any other form of teachings on sexual reproductive health that are happening in many churches that I know about*.(P4)

A young people’s FGD at one church further reported that their “*church is doing nothing to address AYSRH*” (Y2).

Many of the activities were viewed as ineffective and unable to address AYSRH’s needs. The school principals acknowledged that although the “*activities are happening, the quality and effectiveness are very low or poor*” (PT1–PT3). One school principal stated that “*the AYSRH activities in churches are infrequent and unstructured*” (PT2). Another school principal indicated the following:

*The churches’ teaching on sex is still lacking because the pastors once in a while touch on the subject of sex in preaching, but do not discuss it any further. The problem is that the pastors won’t be talking to youth in an environment they are familiar and comfortable, i.e., considering how they think and understand sexual related issues, which happens with their peers openly*.(PT2)

#### 4.2.2. Superficial and Unpractical AYSRH Activities

The respondents indicated that the church’s youth teaching on sex is superficial. It does not provide comprehensive information except for its focus on abstinence. Sometimes, SRH issues are addressed during sermons on Sundays, but there will be no follow-up discussions or the opportunity for anyone to ask questions. One school principal reported the following:

*My church has youth programmes in which they talk to the youth about sex-related issues in passing. I feel it is quite lacking because the conversation about sex during youth programmes does not get to the bottom of the issue*.(PT1)

Hence, one school principal concluded that “churches teach the youth about sexuality, but their teachings are limited” (PT2).

Although church AYSRH services have been implemented, they are not practical or geared towards reality. The overarching message entails (1) abstinence until or unless married; (2) preserving oneself until married; (3) maintaining a clear distinction between the secular world in the message; (4) not seeing sex outside of the realm of God giving it for enjoyment to married people; and (5) viewing AYSRH as synonymous to sex being a sin rather than in health terms. The churches’ teaching is not realistic because young people are sexually involved. As stated by one government official, “*churches’ teaching on sex is very limited. It only focuses on the moral side, without focusing on the health side*” (Gov2).

#### 4.2.3. Tension Between Bible Teachings and Reality: A Reductionistic Approach to AYSRH and Mono AYSRH-Focused Information/Teaching

The participants indicated that there are “murky” and controversial issues regarding AYSRH within churches. The murky issues include teaching on the use of condoms that churches outsource to NGO partners. This perpetually makes churches incapacitated to reconcile this tension by exploring innovative ways to resolve the situation. This dichotomy of the churches’ message on sex and the public health message means that churches will carry on consciously avoiding talking about issues such as condoms to the youth while this task is outsourced to secular NGOs. This keeps churches in their comfort zones, and they continue to be removed from reality.

There is also a reductionist view of AYSRH by churches where they hold only to their one approach of abstinence. Also, SRH is viewed as safe sex by pastors. As indicated by one pastor, “*SRH is safe sex and should focus on abstinence and faithfulness to your married partner over condoms*” (P4). Therefore, this simplistic view needs to be questioned. Churches also focus only on one approach: sharing information or teaching young people about AYSRH-related issues. Teaching and information sharing are the principal methods, but they overlook the development of more realistic skills that should include sex negotiation and other approaches apart from abstinence. Comprehensive AYSRH encompasses far more options than merely providing information. Therefore, a young people’s FGD at TVET posed an important question: *“Churches teach young people only to abstain, which is a narrow view of SRH. They only think of the act of sex and not the other things around it. How could SRH be just one thing?” (YT1)*.

Concern was expressed that while it is acknowledged that the youth are indulging in sex, there is no effort to find ways of realistically engaging with the issue. All pastors admitted to the reality of young people being sexually involved, as clearly reported by one pastor:

*It will be naïve to say that a young Christian person won’t get involved in sexual activity because sex is a temptation. Once a young person has gone past all the barriers set by Christianity, the only hope may be a condom*.(P1)

The message of repentance and abstinence is the single approach held by the church, but it is not effective. A government official commented that “*the Christian message of repentance results in restoration, but there is no message of second virginity. Therefore, churches need to have a message that deals with the reality of youth sex*” (Gov1). Therefore, the view of abstinence as a silver bullet by churches needs to be confronted. One school principal advised the following:

*Comprehensive AYSRH interventions address all the different dimensions and prevent social and health ills such as teenage pregnancy, STIs and HIV. However, the Christian message lacks this holistic view*.(PT2)

#### 4.2.4. Limited Understanding and Lack of Openness to Pastors

The participants reported that pastors lack an in-depth understanding of AYSRH, which fuels tension and conflict in their messages. One pastor admitted that “*churches should operate as a place where youth get reliable information regarding HIV and SRH, but this is not happening*” (P4). This lack of understanding and limited information means that pastors disengage and close themselves off from the AYSRH discussion. Another pastor also admitted the following:


*There is silence in the church about AYSRH, which means that many young people are not receiving the needed direction from the church. The church has gone silent on the matter. It is pointless to be teaching Noah’s ark in Sunday School when your young people are dying of HIV.*


*Some of the signs that we see, particularly in the churches in the Vaal, dare I say to my shame as well, we are not teaching SRH in churches*.(P5)

A further critical dimension pointed out that churches teach abstinence, and the same pastors do not openly speak about sex even when people get married. One parent’ FGD stated the following:

*When people get married in the church, the pastors are the ones who start asking if the wife is pregnant as if they taught them how sex is done. It seems the pastors assume that pregnancy happens automatically*.(PC1)

## 5. Discussion, Study Implications, and Limitations

### 5.1. Discussion

This study reveals that churches across various communities are actively engaging in Adolescent and Youth Sexual Reproductive Health (AYSRH) interventions, utilizing their spaces to offer a range of services and activities. These efforts are often multifaceted, delivered both directly by church members and through collaborations with external stakeholders such as non-governmental organizations, government health institutions, and schools. The approaches encompass a broad spectrum, from imparting moral guidance on sexual conduct and providing individual and group counselling through workshops and youth camps to mentoring young people preparing for marriage and offering resources and information to prevent risky sexual behaviour. Furthermore, churches are employing various methods like highlighting the negative consequences of unsafe sex, establishing youth information centres, training parents to improve their communication with children, and creating mentorship opportunities for parent–child bonding around AYSRH topics. They also organize dedicated youth services and special discussions during camps, provide care and guidance to orphaned youth on sexual matters within a religious framework, and participate in national events to address issues like teenage pregnancies.

Despite these commendable efforts, the implementation of AYSRH activities within churches is not without its challenges and areas of concern. Some respondents in the study reported a lack of AYSRH services altogether in certain churches, while in others, the existing services were deemed neither meaningful nor effective. Concerns were raised that the interventions could be superficial, merely touching on the surface of issues without in-depth follow-up and that the information provided might be impractical and disconnected from the lived experiences of young people. Furthermore, a perceived dichotomy between church-based AYSRH services and those offered by government and international organizations was noted, potentially hindering partnerships. Critics also pointed out a tendency towards a reductionistic approach that primarily emphasizes abstinence and sex within marriage, often focusing solely on biblical interpretations of morality. The study also highlighted an apparent inability to navigate the conflict between religious ideals and the realities of young people’s sexual involvement, as well as silence or a lack of clarity on crucial topics like the effectiveness of condoms within Christian messaging. Moreover, the interventions were often criticized for relying on a single approach centred on religious teachings, coupled with a limited understanding of AYSRH among some pastors, which can lead to a lack of openness and uncertainty in their engagement with these sensitive issues, ultimately resulting in a narrow view of AYSRH primarily focused on sex and sexuality.

The study provides evidence confirming the differences both between and within faith-based organizations. It also supports findings from other AYSRH studies, which highlight the significant variations among churches and faith communities, making it difficult to draw a single conclusion [21,32,33,34]. Some churches have developed advanced partnerships and resources, such as manuals, while others lack AYSRH materials or activities entirely. This diversity within churches and broader faith organizations is aptly captured by Wilkinson et al. [34], who noted the following:


*Faith actors, just like other political and social actors, are incredibly diverse and represent the full gamut of possible opinions. This is not only a question of diversity across different religions, but diversity within denominations and schools, and diversity in opinions down to individuals in a local faith community.*


The variety of implemented interventions and the presence of churches with no activities at all reflect this heterogeneity. This study highlights both the AYSRH services implemented by churches and the differences in these interventions, providing crucial insights into the nuances of congregational AYSRH implementation. This contributes to understanding the realities at the grassroots level of churches, which is an area often overlooked. Research indicates that health and development practitioners tend to engage with top-level faith leaders while lacking an understanding of the grassroots level. Wilkinson et al. [34] emphasize the importance of localized and in-depth research to uncover contextual nuances. By showcasing the limited services offered and their shortcomings, this study provides a clearer picture of the realities of AYSRH implementation at the congregational level. It moves beyond the essentialist notion of what churches “should” be doing to a more realistic perspective on what they are actually doing, revealing both opportunities and challenges.

However, this study also found that church approaches remain predominantly focused on abstinence until marriage, with resistance to promoting condom use, which is deemed unrealistic. This position emerged across various church programmes and communication platforms, including youth groups, gender-based groups, camps, and sermons. On a positive note, churches emphasize biblical and moral messaging, aligning with their spiritual mission. Their primary role in AYSRH interventions is information dissemination, often embedded in traditional activities such as catechism, discipleship, and Bible studies. Religious leaders wield influence through their messages and the respect they command [35,36,37,38]. While the singular focus on communication is a limitation—such as preaching without considering social realities—it also offers predictability when establishing partnerships with churches.

This study identifies a dichotomy in the church’s approach to health, dividing its operations into spiritual and practical aspects. On the practical side, churches manage hospitals and schools, while their spiritual work is centred on clergy duties. This division often results in inconsistencies between their management of healthcare institutions and their spiritual teachings [39]. Consequently, there is a limited understanding of health interventions at the “congregational ecology level” [39]. This study sheds light on the congregational ecology of health services, complementing existing research on churches at the grassroots level [23,40].

Church interventions lack diversity and fail to align with comprehensive AYSRH frameworks. The Universal Declaration of Human Rights, the International Conference on Population and Development (1994), UNFPA, and the WHO define AYSRH as encompassing physical, emotional, mental, and social well-being in all aspects of sexuality and reproduction. ASRH interventions aim not just to prevent disease but to empower adolescents with rights-based access to services [34].

The WHO and Guttmacher-Lancet Commission recommend an integrated AYSRH approach that includes promotion, prevention, and curative interventions while addressing social determinants of health [41]. These include comprehensive education on sexuality, contraceptive services, antenatal and postnatal care, safe abortion services, STI prevention and treatment, GBV responses, reproductive cancer management, infertility support, and sexual health counselling. AYSRH services are categorized into demand generation (encouraging service uptake) and supply-side interventions (strengthening service provision in health facilities) [42,43,44].

Based on these frameworks, church interventions fall short. Their focus on abstinence and moral messaging is inadequate in addressing the evolving realities of AYSRH. However, criticism of churches must be tempered with realistic expectations. Unlike NGOs and government institutions, churches prioritize faith-based activities. Expecting them to implement comprehensive AYSRH interventions is unrealistic, especially given that many pastors admit to having “little knowledge of AYSRH” due to the absence of sex-related topics in theological training. Nevertheless, church leaders perform public roles, such as raising AYSRH awareness in schools. Public theology has emerged as a means to engage churches with social issues, including AYSRH [45].

Pastors already juggle multiple responsibilities, including visiting the sick, organizing youth groups, leading Bible classes, and officiating ceremonies. Expecting them to implement further AYSRH interventions—without medical, sociological, or psychological training—is impractical. The key question is the following: how can pastors effectively contribute to AYSRH? Despite criticism of their reductionist approaches, the strength of churches lies in their capacity to share information, while partnerships can fill gaps in service provision. Both pastors and stakeholders acknowledge that churches rely on partnerships to implement AYSRH interventions.

Churches are already engaged in the generation of demand through AYSRH information dissemination in various settings, but they do not contribute to the supply side. Strengthening their communication efforts while expanding partnerships with healthcare providers could enhance their role. This study identifies existing services, highlights weaknesses, and presents opportunities for strengthening AYSRH interventions through collaborative approaches.

### 5.2. Study Implications

This study has three key implications. First, the variations among faith-based organizations must be carefully studied when developing church-based AYSRH interventions. These variations exist at multiple levels, including intra-congregational, inter-congregational (within the same or different denominations), intra-denominational, and inter-denominational levels and across geographical and social contexts. Understanding these dynamics can ensure that interventions are contextually relevant rather than imposed through top-down approaches.

Second, this study affirms the importance of a religious health asset (RHA) approach while identifying pastors’ limited knowledge of AYSRH as a critical gap. RHA focuses on leveraging strengths rather than deficits, with churches excelling in information-sharing. Training pastors on AYSRH would enable them to operate more effectively within their areas of strength.

Third, this study presents two viable options for churches in AYSRH implementation. They can either utilize their internal resources and leadership strengths or establish partnerships to address areas where they lack competence. Strengthening partnerships with healthcare providers and NGOs could bridge gaps in service delivery.

### 5.3. Study Limitation

This study examined AYSRH services from pastors’ perspectives (emic) and external stakeholders’ viewpoints (etic). This revealed a divergence. Pastors defended their approach, while stakeholders criticized churches for their shortcomings. Future research should employ a mixed-method approach with a larger sample to explore these contrasting perspectives further.

Acknowledging the inherent constraints of an exploratory research design exclusively reliant on qualitative data, a recognized limitation of this study was the absence of a complementary quantitative component. Such a component would have afforded the researcher the capacity to triangulate the findings and establish statistical significance, thereby enhancing the generalizability and robustness of the conclusions. To address this methodological limitation and to compensate for the lack of numerical data, a significantly enlarged qualitative sample (n = 24) from diverse participants was strategically employed (i.e., KII = 16; FGDs: eight). This augmentation aimed to maximize the depth and breadth of the qualitative data, providing a more comprehensive understanding of the phenomenon under investigation. While not fully replicating the analytical power of a mixed-methods approach, the utilization of a substantial qualitative sample sought to bolster the study’s credibility and mitigate the potential for interpretive bias, ultimately contributing to a more nuanced and defensible analysis.

## 6. Conclusions

This study investigated AYSRH interventions implemented by churches, examining their alignment with religious beliefs and practices to identify strategies for enhancing intervention effectiveness. The research identified several distinct AYSRH activities, primarily centred on information dissemination and the promotion of abstinence until marriage, grounded in the moral values of chastity. These interventions were delivered through two primary mechanisms: pastor-led services integrated into church programmes and collaborative partnerships with external stakeholders. These partnerships facilitate the delivery of interventions addressing controversial topics, such as condom use and family planning, which fall outside pastors’ perceived areas of competence. This study highlights information and communication platforms, as well as established partnerships, as key strengths of churches that can be leveraged for AYSRH initiatives. While confirming findings from existing research on church-based AYSRH, this study provides nuanced insights into the one-sided focus of church teachings and exposes the tensions between idealized approaches and practical implementation. This tension raises significant questions about the overall efficacy of church-led AYSRH projects.

## Figures and Tables

**Figure 1 healthcare-13-00907-f001:**
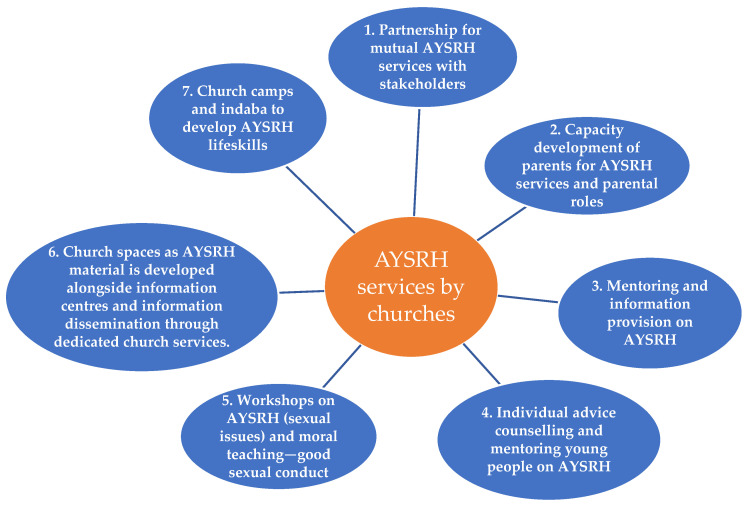
AYSRH services by churches.

**Figure 2 healthcare-13-00907-f002:**
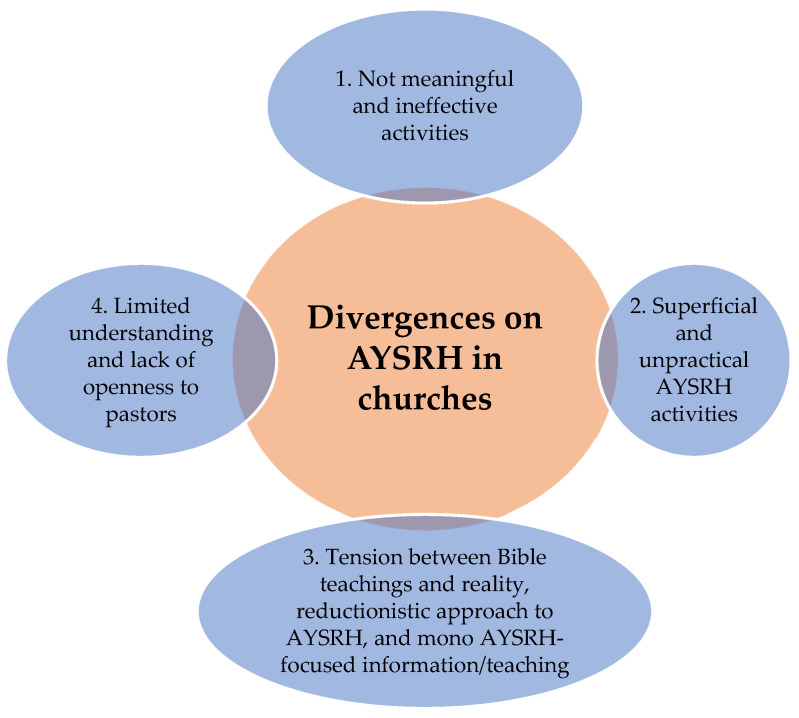
Divergences on AYSRH in churches.

**Table 1 healthcare-13-00907-t001:** List of samples, numbers, and assigned pseudonyms for the three data collection sites.

Vaal Area	List of Pastors	Pseudonym Assigned
**Pastors (11) In-Depth Interviews (II) (VDBK–6; VR–2; Sasol–3)**		
Vanderbijlpark	Pastor 1	P1
Sasolburg	Pastor 2	P2
Vanderbijlpark	Pastor 3	P3
Vanderbijlpark	Pastor 4	P4
Vereeniging	Pastor 5	P5
Sasolburg	Pastor 6	P6
Vanderbijlpark	Pastor 7	P7
Vanderbijlpark	Pastor 8	P8
Vereeniging	Pastor 9	P9
Sasolburg	Pastor 10	P10
Vanderbijlpark	Pastor 11	P11
**Parents FGDS (VDBK–1; VR–1; Sasol–1)**		
Vanderbijlpark	Church 1	PC1
Vereeniging	Church 2	PC2
Sasolburg	Church 3	PC3
**School Principals (VDBK–1; VR–1; Sasol–1)**		
Vanderbijlpark	Principal 1	PT1
Sasol	Principal 2	PT2
Vereeniging	Principal 3	PT3
**Government Officials In-depth Interviews (2) (District Officials DSD–1; DoH–1)**		
Sedibeng District	Department of Social Development	GOV1
Sedibeng District	Department of Health	GOV2
**Young people Church FGDs (VR–1; VDBK–1; Sasol–1)**		
Sasolburg	Church 1	Y1
Vereeniging	Church 2	Y2
Vanderbijlpark	Church 3	Y3
**Young people TVET FGDs (VR–1; VDBK–1)**		
Vanderbijlpark	Sedibeng TVET Vanderbijlpark Campus	YT1
Vereeniging	Sedibeng TVET Vereeniging Campus	YT2

## Data Availability

The original contributions presented in this study are included in the article. Further inquiries can be directed to the corresponding author.

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
