# Peer review of "Understanding Church-Led Adolescent and Youth Sexual Reproductive Health (AYSRH) Interventions Within the Framework of Church Beliefs and Practices in South Africa: A Qualitative Study"

_healthcare, 2025, doi:10.3390/healthcare13080907_

Round 1
Reviewer 1 Report
Comments and Suggestions for Authors
The study addresses a crucial concern in adolescent and young population's health, namely sexual and reproductive health. While it starts out promising, it needs improvement of the methodological explanation and, more importantly, of the discussion of findings which makes out ca. 60% of the entire paper. Please see attached file for more exact suggestions for improvement.

Author Response
Comment 1: Editorials and document cleaning
Response: This has been done throughout the document Done
Comment 2: Consistent use of VDBK
Response: This has been where VDBK is mentioned (see table 1) line 279ff
Comment 3: Findings figure 1 & subsequent discussion difficult to follow.
Response: The figure has been numbered and discussion aligned to the numbering for easy following. See lines 341-582)
Findings – section on divergences in AYSRH confusing like figure 1.
Figure 2 has been introduced, number and discussion aligned to the numbering on figure for easy following. See lines 583-697.

Reviewer 2 Report
Comments and Suggestions for Authors
This study aims to inform interventions aimed at improving adolescent and youth sexual reproductive health (AYSRH), using the insights from the church. The authors should consider improving the description of the purpose, theoretical framework, and methods so that they align well.
- Also, the authors may want to improve readability. To exemplify, the opening sentence of the abstract comprises 72 words and 456 characters. To improve readability, similar issues in writing may be addressed by breaking those complex sentences into several. The authors should address the following specific comments to improve the clarity of this article.
- Title: The title will benefit from highlighting the “qualitative” research focus. The title is also misguiding in that a reader is not sure what kind of interventions can benefit from the article. It is likely that large-scale public health interventions may not be able to use these insights, but other churches can. Perhaps adding “church-led” in the title will address this issue. It will read as “Towards understanding church-led Adolescent and Youth Sexual Reproductive Health (AYSRH) ….”
Abstract:
- The abstract should include some methodological details critical to understanding the rigor—specifying the analysis approach and software used for analysis is important.
Introduction:
- The Introduction section needs better alignment and flow of ideas. This section lacks depth (more studies need to be cited to support the argument). The authors should consider elaborating several key elements that would strengthen the study’s foundation. For instance, a mention is made of the theoretical frameworks used in such study without adequately describing specific theoretical framework(s) that guided the conceptualization of the study. Merging the introduction section with the literature review section will improve the Introduction and eliminate redundancies between these two sections.
- Line 71-73: This study claims “to explore AYSRH interventions implemented by churches that are compatible with their faith beliefs and values” but provided theoretical information rather than mentioning why this study was conducted. The purpose of the study is alluded to at two different places in the Introduction. The authors should consider clearly providing justification for the study by establishing and describing research gaps, specifically making the research useful.
- Line 39: The authors should cite the source for the sentence, “There is a significant lack of empirical research on practical church-based AYSRH interventions.” However, the study by Van Bortel et al. (2019) is mentioned after this sentence, but an additional supporting reference should be provided.
- The authors have not justified why a “literature Review” as a separate section is needed. By integrating the literature review with the Introduction section, the authors can address the comments about the lack of depth in the instruction.
Methodology:
- Among the critical gaps in the methodology include the absence/inadequacy of the description of the analysis process. The authors are expected to describe which software was used for the thematic analysis, who was involved in the thematic coding, how intercoder reliability was checked, and what considerations guided the coding. The absence of such details makes it difficult to establish whether the results are unbiased.
- The authors should consider reorganizing the methodology sections. They should mention how the "in-depth interviews and focus groups" were conducted, such as the duration of the interviews and focus groups, the type of questions, the interview guide, or themes.
- Although ethical approval (NWU-00879-19-S7) is mentioned, the details on informed consent from participants or a statement confirming their voluntary participation are missing. The author should consider detailing how the participants were approached and whether anonymity and confidentiality measures were followed.
- Line 139: The authors should discuss the coding process, how many themes were identified/verified, whether any software was used, and the details of any reliability of measures taken during the coding procedure.
- Line 152-158: The author should provide a rationale for the sample selected. Why only TVET students, government officials, and churchgoers? The details of whether the type of sampled study participants have any specific role in the church-guided interventions should be justified (perhaps in the Introduction). How was the potential selection bias checked/addressed?
- Table 1 has abbreviations for the terms used in the study. The author should mention the complete form at the end of the table in the footnotes.
Results
- The results are confusing and need clarity regarding the role of churches in providing the necessary services and meaningful interventions. The author should mention whether churches offer services. For example, some pastors state that churches provide AYSRH interventions (Lines 180–200), while others claim nothing meaningful is done (Lines 459–472).
- (Lines 176–195 vs. 367–372): The authors seem to find it challenging to contextualize divergent views. For instance, the findings include that churches are safe places and provide mentoring programs. However, the findings also say, "This relationship enables some stakeholders to complement the churches' AYSRH information through addressing topics that churches are uncomfortable dealing with."
- Some participants stated that church NGOs actively work for AYSRH partnership programs (Lines 205–225), while others reported that churches discourage topics NGOs handle (Lines 514–519).
Discussion
- A critical flaw in the discussion of the results is a lack of sufficient description of the study limitations. This amounts to a lack of transparency. It also is a missed opportunity to demonstrate the positionality and reflexivity of the authors, which are desirable for qualitative studies.
- Discussion of the current study results is meaningful within the context of previous studies and their findings. The author should add studies that support the findings by addressing counterarguments and contradictions.
- There is confusion regarding the church authorities' decision to expand their role or focus on partnerships only. The authors should provide supporting evidence for lines 634–635, where pastors need training. Also, the author should provide supporting data for the churches' insufficient supply-side interventions (Lines 646–647). The authors should add relevant research studies to support the study findings into the discussion.
- References: The in-text and listed references do not conform to the journal requirements.
The authors should improve readability. To exemplify, the opening sentence of the abstract comprises 72 words and 456 characters. To improve readability, similar issues in writing may be addressed by breaking those complex sentences into several.
Author Response
Comments group 1:
- Also, the authors may want to improve readability. To exemplify, the opening sentence of the abstract comprises 72 words and 456 characters. To improve readability, similar issues in writing may be addressed by breaking those complex sentences into several. The authors should address the following specific comments to improve the clarity of this article.
- Title: The title will benefit from highlighting the “qualitative” research focus. The title is also misguiding in that a reader is not sure what kind of interventions can benefit from the article. It is likely that large-scale public health interventions may not be able to use these insights, but other churches can. Perhaps adding “church-led” in the title will address this issue. It will read as “Towards understanding church-led Adolescent and Youth Sexual Reproductive Health (AYSRH) ….”
Responses
1. Readability has been improved by shortening sentences (see abstract). Lines 10-34
2. The title has been revised to incorporate reviewer comments to read as follows “Towards understanding church-led Adolescent and Youth Sexual Reproductive Health (AYSRH) interventions within the framework of church beliefs and practices in South Africa: a qualitative study” (lines 1-3).
Abstract
Comment
- The abstract should include some methodological details critical to understanding the rigor—specifying the analysis approach and software used for analysis is important.
Response: Done. Additional detail including software analysis software has been provided (see lines 18-20) on abstract.
Introduction:
Comments group 2 introduction
- The Introduction section needs better alignment and flow of ideas. This section lacks depth (more studies need to be cited to support the argument). The authors should consider elaborating several key elements that would strengthen the study’s foundation. For instance, a mention is made of the theoretical frameworks used in such study without adequately describing specific theoretical framework(s) that guided the conceptualization of the study. Merging the introduction section with the literature review section will improve the Introduction and eliminate redundancies between these two sections.
- Line 71-73: This study claims “to explore AYSRH interventions implemented by churches that are compatible with their faith beliefs and values” but provided theoretical information rather than mentioning why this study was conducted. The purpose of the study is alluded to at two different places in the Introduction. The authors should consider clearly providing justification for the study by establishing and describing research gaps, specifically making the research useful.
- Line 39: The authors should cite the source for the sentence, “There is a significant lack of empirical research on practical church-based AYSRH interventions.” However, the study by Van Bortel et al. (2019) is mentioned after this sentence, but an additional supporting reference should be provided.
- The authors have not justified why a “literature Review” as a separate section is needed. By integrating the literature review with the Introduction section, the authors can address the comments about the lack of depth in the instruction.
Responses
· The introduction has been reformulated. The aim of the study is clearly indicated as exploratory without a specific single theoretical framework. (see lines 39-124)
· The rationale/justification has been provided in the background section that combined the previous introduction and literature discussion.
· (see lines 91-99, 111-124)
· References have been better cited and aligned to avoid referencing confusion/misunderstanding. All sources have been cited.
· The section on literature review has been integrated into background discussion and made to flow logically (see lines 39-124).
Methodology:
Comments
- Among the critical gaps in the methodology include the absence/inadequacy of the description of the analysis process. The authors are expected to describe which software was used for the thematic analysis, who was involved in the thematic coding, how intercoder reliability was checked, and what considerations guided the coding. The absence of such details makes it difficult to establish whether the results are unbiased.
- The authors should consider reorganizing the methodology sections. They should mention how the "in-depth interviews and focus groups" were conducted, such as the duration of the interviews and focus groups, the type of questions, the interview guide, or themes.
- Although ethical approval (NWU-00879-19-S7) is mentioned, the details on informed consent from participants or a statement confirming their voluntary participation are missing. The author should consider detailing how the participants were approached and whether anonymity and confidentiality measures were followed.
- Line 139: The authors should discuss the coding process, how many themes were identified/verified, whether any software was used, and the details of any reliability of measures taken during the coding procedure.
- Line 152-158: The author should provide a rationale for the sample selected. Why only TVET students, government officials, and churchgoers? The details of whether the type of sampled study participants have any specific role in the church-guided interventions should be justified (perhaps in the Introduction). How was the potential selection bias checked/addressed?
- Table 1 has abbreviations for the terms used in the study. The author should mention the complete form at the end of the table in the footnotes.
Responses
· A detailed methodology section has been included that addresses all the different aspects raised. See two sections on methodology (see lines 136-210).
· A detailed analysis description of the process has been provided (see lines 213-280)
· The expanded methodology section includes the details on interviews duration, type of questions in the interview guide (see lines 175-179).
· Details on ethical process have been provided (see lines 180-193).
· The coding process followed have been provided (see lines 207-219). However, to avoid the methodology to be too long, the details on themes from the coding have been excluded. The methodology section has become too long, which risk discouraging the reader looking for results.
· Rationale for sample justification has been provided, see lines 142-158.
· Abbreviations in table have been corrected (see line 279 ff).
Results
Comment
- The results are confusing and need clarity regarding the role of churches in providing the necessary services and meaningful interventions. The author should mention whether churches offer services. For example, some pastors state that churches provide AYSRH interventions (Lines 180–200), while others claim nothing meaningful is done (Lines 459–472).
Response
· Results and discussion issues raised have been attended as advised. See respective section.
· The difference in pastors’ responses regarding services have been discussed as divergences and tensions as a key finding. A diagram to make this clear figure 2 has been provided (see 587 -588 & figure 2).
· The divergences have been discussed in a divergence and tensions section (see 587 -588 & figure 2).
Comments
- (Lines 176–195 vs. 367–372): The authors seem to find it challenging to contextualize divergent views. For instance, the findings include that churches are safe places and provide mentoring programs. However, the findings also say, "This relationship enables some stakeholders to complement the churches' AYSRH information through addressing topics that churches are uncomfortable dealing with."
- Some participants stated that church NGOs actively work for AYSRH partnership programs (Lines 205–225), while others reported that churches discourage topics NGOs handle (Lines 514–519).
Response
The difference in pastors’ responses regarding services have been discussed as divergences and tensions as a key finding. A diagram to make this clear figure 2 has been provided (see 587 -588 & figure 2).
The divergences have been discussed in a divergence and tensions section (see 587 -588 & figure 2).
Discussion
Comment
- A critical flaw in the discussion of the results is a lack of sufficient description of the study limitations. This amounts to a lack of transparency. It also is a missed opportunity to demonstrate the positionality and reflexivity of the authors, which are desirable for qualitative studies.
Response
The study limitations have been provided. And on methodology section, the process followed to ensure rigor have also been provided (see lines 237-278).
Comment
- Discussion of the current study results is meaningful within the context of previous studies and their findings. The author should add studies that support the findings by addressing counterarguments and contradictions.
Response
The studies supporting the findings have been provided in the discussion. For instance, see in text references (e.g. lines 739-780) - Cense et al., 2018; Koletić et al., 2021; Li et al., 2016; Mbarushimana et al., 2022; Powell et al., 2017; Wilkinson et al., 2019, Greyling et al., 2016; Magezi, 2018, Health Action International, 2022; Jajkowicz, 2014; Stawski, 2012; UNICEF, 2012)
Comment
- References: The in-text and listed references do not conform to the journal requirements.
Response: See corrected reference list (lines 874-1099)

Reviewer 3 Report
Comments and Suggestions for Authors
Attached are my comments.

Author Response
Comment: 1. Methodological Scope – The sample size is relatively small (11 pastors, 2 government officials, 3 parent groups, and 3 youth groups). Expanding the participant pool or incorporating quantitative data could enhance the study’s generalizability.
Response: Study limitations have been clearly described and mitigation. See relevant sections.
- Comment: Critical Engagement with Abstinence-Based Approaches – The study acknowledges that churches primarily promote abstinence until marriage. However, it could further discuss the effectiveness of abstinence-based strategies compared to comprehensive sexual education, particularly in reducing unintended pregnancies and sexually transmitted infections.
Response: To the extent possible, the study discusses the issues, but discussion was informed by emerging themes. See discussion section.
3.Comment: Intersectionality and Inclusivity – The study could explore whether church-based interventions adequately address the needs of marginalized youth, including LGBTQ+ individuals, young mothers, or adolescents from different socio-economic backgrounds.
Response: The recommended areas will be explored in future studies.
- Comment 4: Long-Term Impact and Outcomes – While the research outlines current church interventions, it does not seem to evaluate their long-term effectiveness. Including follow-up data or comparative case studies would strengthen the findings.
Response: Certainly, future studies currently underway will address the recommendations.
5.Comment: Policy Implications – The discussion could better connect findings to policy recommendations, suggesting ways churches and government agencies could collaborate for more inclusive and effective AYSRH interventions.
Response: This will be a focus in the next papers.
Comment: Moreover, the study could benefit if the writers address the following issues:
- The study prioritizes pastors' and stakeholders' views but lacks input from young beneficiaries.
Response: The views of young beneficiaries is provided by young people in churches and Colleges through FGDs.
- A comparison with secular or government-led initiatives would provide a broader context.
Comment: Need for Concrete Recommendations:
Response: These have been provided in the implications sections.

Round 2
Reviewer 2 Report
Comments and Suggestions for Authors
The authors have done a good job in addressing my comments from report 1.